# *n*-3 and *n*-6 Polyunsaturated Fatty Acids Modulate Macrophage–Myocyte Inflammatory Crosstalk and Improve Myocyte Insulin Sensitivity

**DOI:** 10.3390/nu16132086

**Published:** 2024-06-29

**Authors:** Amber L. Hutchinson, Danyelle M. Liddle, Jennifer M. Monk, David W. L. Ma, Lindsay E. Robinson

**Affiliations:** Department of Human Health and Nutritional Sciences, University of Guelph, Guelph, ON N1G 2W1, Canadajmonk02@uoguelph.ca (J.M.M.); davidma@uoguelph.ca (D.W.L.M.)

**Keywords:** myocytes, macrophages, inflammation, fatty acids, insulin, obesity

## Abstract

In obesity, circulating saturated fatty acids (SFAs) and inflammatory cytokines interfere with skeletal muscle insulin signaling, leading to whole body insulin resistance. Further, obese skeletal muscle is characterized by macrophage infiltration and polarization to the inflammatory M1 phenotype, which is central to the development of local inflammation and insulin resistance. While skeletal muscle-infiltrated macrophage–myocyte crosstalk is exacerbated by SFA, the effects of other fatty acids, such as *n*-3 and *n*-6 polyunsaturated fatty acids (PUFAs), are less studied. Thus, the objective of this study was to determine the effects of long-chain *n*-3 and *n*-6 PUFAs on macrophage M1 polarization and subsequent effects on myocyte inflammation and metabolic function compared to SFA. Using an in vitro model recapitulating obese skeletal muscle cells, differentiated L6 myocytes were cultured for 24 h with RAW 264.7 macrophage-conditioned media (MCM), followed by insulin stimulation (100 nM, 20 min). MCM was generated by pre-treating macrophages for 24 h with 100 μM palmitic acid (16:0, PA–control), arachidonic acid (20:4*n*-6, AA), or docosahexaenoic acid (22:6*n*-3, DHA). Next, macrophage cultures were stimulated with a physiological dose (10 ng/mL) of lipopolysaccharide for an additional 12 h to mimic in vivo obese endotoxin levels. Compared to PA, both AA and DHA reduced mRNA expression and/or secreted protein levels of markers for M1 (TNFα, IL-6, iNOS; *p* < 0.05) and increased those for M2 (IL-10, TGF-β; *p* < 0.05) macrophage polarization. In turn, AA- and DHA-derived MCM reduced L6 myocyte-secreted cytokines (TNFα, IL-6; *p* < 0.05) and chemokines (MCP-1, MIP-1β; *p* < 0.05). Only AA-derived MCM increased L6-myocyte phosphorylation of Akt (*p* < 0.05), yet this was inconsistent with improved insulin signaling, as only DHA-derived MCM improved L6 myocyte glucose uptake (*p* < 0.05). In conclusion, dietary *n*-3 and *n*-6 PUFAs may be a useful strategy to modulate macrophage–myocyte inflammatory crosstalk and improve myocyte insulin sensitivity in obesity.

## 1. Introduction

Obesity is defined by low-grade inflammation that drives the development of chronic metabolic diseases [1,2]. In skeletal muscle, the primary site of insulin-stimulated glucose disposal [3,4], the infiltration of certain immune cells and their paracrine interactions (crosstalk) with myocytes, and the ectopic lipid accumulation characteristic of obesity as a result of adipose tissue (AT) dysfunction contribute to the development of local and systemic insulin resistance [4,5,6,7]. Mechanistically, inflammatory cytokines and SFA impair insulin signaling through reduced serine phosphorylation and activation of Akt as well as increased threonine phosphorylation of the insulin receptor substrate 1 (IRS-1), resulting in reduced translocation of the glucose transporter type 4 (GLUT4) to the plasma membrane and reduced glucose uptake [5,8,9,10]. Hence, inflammation is a potential target to mitigate the development of skeletal muscle insulin resistance and the ensuing consequences in obesity.

M1 macrophages are the hallmark of inflammation and insulin resistance in obesity [11]. Monocytes are recruited to obese skeletal muscle and differentiate into macrophages in response to increased myocyte production of monocyte chemoattractant 1 (MCP-1), as demonstrated in experiments of MCP-1 overexpression in myocytes in vitro and in high-fat diet (HFD)-fed MCP-1 knockout rodent models in vivo [12,13]. Macrophages are polarized to the M1 phenotype and are activated in response to stimulation with the cytokine interferon (IFN)-γ and toll-like receptor (TLR)4 agonists, including SFA, such as palmitic acid (16:0; PA), and lipopolysaccharide (LPS), which act to (i) phosphorylate and activate nuclear transcription factors nuclear factor-κB (NF-κB) and signal transducer and activator of transcription 1 (STAT1), (ii) increase the expression of iNOS, and (iii) stimulate the expression of inflammatory cytokines, including TNFα and IL-6 [14,15]. M1 macrophage-secreted inflammatory cytokines contribute to macrophage–myocyte crosstalk, which further perpetuates the inflammatory microenvironment within obese skeletal muscle [14].

Macrophage–myocyte inflammatory crosstalk provides a potential target for intervention with anti-inflammatory nutrients, including dietary long-chain polyunsaturated fatty acids (PUFAs). The *n*-3 PUFAs, eicosapentaenoic acid (20:5*n*-3, EPA) and docosahexaenoic acid (22:6*n*-3, DHA), have well-known anti-inflammatory effects in obesity via attenuating skeletal muscle inflammation [16,17,18] and LPS-induced M1 macrophage polarization [19,20], with DHA being more potent than EPA in the latter context [19]. The effects of the *n*-6 PUFA, arachidonic acid (20:4n-6, AA), are generally considered to be inflammatory due to the production of AA-derived metabolites from the cyclooxygenase (COX) pathway [21]. However, controversy exists, as macrophages treated with AA and its metabolite PGE_2_ have been shown to inhibit LPS-induced inflammation and promote expression of markers for the anti-inflammatory M2 polarization [22,23,24]. While *n*-6 and *n*-3 PUFAs have been studied in obese skeletal muscle and macrophages separately, their effect on macrophage polarization and subsequent myocyte function are unknown. Hence, using an in vitro model that mimics the obese skeletal muscle inflammatory microenvironment, the objective of the current study was to determine the effects of the long-chain *n*-6 PUFA, AA, and *n*-3 PUFA, DHA, on macrophage M1 polarization and the subsequent effects on myocyte inflammation and metabolic function compared to the SFA, PA.

## 2. Materials and Methods

### 2.1. Cell Culture and Differentiation

RAW 264.7 murine macrophages (ATCC, American Type Culture Collection (TIB-71), Manassas, VA, USA) and L6 rat myoblasts (ATCC (CRL-1458)) were grown and passaged according to the manufacturer’s instructions. In brief, both cell types were maintained in basic media: Dulbecco’s Modified Eagles Medium (DMEM) containing 4500 mg/L glucose, 4.0 mM L-glutamine, and 1 mM sodium pyruvate (HyClone, Logan, UT, USA) and supplemented with 10% (*v*/*v*) fetal bovine serum (FBS; low-endotoxin, sterile-filtered, Millipore-Sigma, Oakville, ON, Canada) and 1% (*v*/*v*) penicillin-streptomycin (HyClone) in a humidified incubator at 37 °C in 5% carbon dioxide. RAW 264.7 macrophages were seeded into 6-well plates at 1 × 10^6^ cells/mL, and on day 2, the media was replaced with serum-reduced media (0.25% FBS) for 12 h prior to treatment. L6 myoblasts were seeded into 6-well plates at 4 × 10^4^ cells/mL. At 70–80% confluence, cells were differentiated and maintained as mature myocytes using differentiation media supplemented with 2% (*v*/*v*) FBS and 1% (*v*/*v*) penicillin-streptomycin. Media was changed on days 2, 4, and 6, and on day 7 the media was replaced with serum-reduced media (0.25% FBS) for 12 h prior to treatment on day 8.

### 2.2. RAW 264.7 Macrophage Treatment and Macrophage–Myocyte Co-Culture Conditions

Fatty acids (FA; Cayman Chemical, Ann Arbor, MI, USA) were prepared to equimolar stock solutions in lab-grade ethanol, and the bovine serum albumin (BSA; endotoxin-free, FA-free; Millipore-Sigma) stock solution was diluted to 20 μM in serum-reduced media. Treatments were prepared by complexing 100 μM of palmitic acid (16:0; PA), arachidonic acid (20:4*n*-6; AA), or docosahexaenoic acid (22:6*n*-3; DHA) with BSA in serum-free DMEM at a 5:1 molar ratio (i.e., 100 μM FA:20 μM BSA), as previously described [25]. Lipopolysaccharide (LPS) from *Escherichia coli* 055:B5 (Millipore-Sigma) was dissolved in serum-deprived media (0% FBS and 1% (*v*/*v*) penicillin-streptomycin) and diluted to a final concentration of 10 ng/mL prior to the addition to treatment media. This LPS dose recapitulates the circulating levels of LPS reported in obese humans [26] and rodents [27,28]. Macrophages were treated with FA for 24 h followed by stimulation with LPS for 12 h (i.e., 36 h total). The macrophage cell culture supernatant (macrophage conditioned media, MCM) was collected, pooled within FA treatments, and stored at −80 °C until it was used to treat L6 myocytes. MCM was thawed and filtered prior to treating L6 myocytes for 24 h. L6 myocytes were treated with MCM for 24 h followed by serum-deprived media for 3–5 h prior to stimulation with a maximal dose of insulin (100 nM) for 20 min.

### 2.3. Cellular Fatty Acid Composition

Total lipids were extracted and processed from 2 × 10^6^ RAW 264.7 macrophages (*n* = 4–5/treatment) as described previously [29], with minor modification. FA methyl esters were separated by gas chromatography using an Agilent 6890N gas chromatograph (Agilent Technologies, Santa Clara, CA, USA). FA peaks were identified by comparing retention times of the samples with those of known standards (Nu-Chek-Prep Inc., Elysian, MN, USA) and analyzed using EZchrom Elite version 3.2.1. software (Agilent Technologies).

### 2.4. mRNA Expression Analysis

RAW 264.7 macrophages (*n* = 7/treatment) and L6 myocytes (*n* = 5–6/treatment) were lysed after treatment to isolate RNA and protein using the RNA/protein purification kit as per the manufacturer’s instructions (Norgen Biotek Corp., Thorold, ON, Canada). cDNA was synthesized using a high-capacity cDNA reverse transcription kit (Applied Biosystems, Foster City, CA, USA) and real time (RT)-PCR was performed using the 7900HT Fast RT-PCR system (Applied Biosystems), as previously described [30]. Primer sequences are shown in Appendix A for markers of M1 (*iNos*, *Cd11c*) and M2 (*Arg-1*, *Cd206*) macrophage polarization and inflammatory (*Tnfα, Il-6*, *Il1β*) and anti-inflammatory (*Il10*, *Tgfβ*) cytokines as well as chemokines (*Mcp-1*). Results were normalized to the expression of the housekeeping genes, 18S for RAW 264.7 macrophages and ribosomal protein, large, P0 (Rplp0) for L6 myocytes, and the relative differences in gene expression between groups were determined according to the calculation 2^(40-Ct)^.

### 2.5. Analysis of NF-κB and STAT3 Activation

RAW 264.7 macrophage and L6 myocyte total cellular protein was quantified using the bicinchoninic assay (Fisher Scientific, Mississauga, ON, Canada), and an equal amount of protein (10 mg/sample/assay) was utilized for the following InstantOne ELISA kits (eBioscience, San Diego, CA, USA), according to the manufacturer’s instructions. Phosphorylated (i.e., activated) NF-κB p65 (S536) and STAT3 (Y701) was measured in RAW 264.7 macrophages treated with FA for 24 h followed by LPS stimulation for 12 h (*n* = 6–7/treatment). Phosphorylated NF-κB p65 was also measured in L6 myocytes following treatment with MCM for 24 h (*n* = 6–8/treatment).

### 2.6. Secreted Protein Analysis

Supernatant was collected from RAW 264.7 macrophages (*n* = 7–9/treatment) following pre-treatment with FA (24 h) and subsequent LPS stimulation (12 h) for measurement of secreted cytokines (MCP-1, TNFα, IL-6, and IL-10) using a ProcartaPlex Mouse Basic kit (eBioscience). The supernatant from L6 myocytes (*n* = 7–8/treatment) following 24 h treatment with MCM was collected for measurement of combined macrophage and L6 myocyte secreted cytokines (TNFα, IL-6) and chemokines (MCP-1, MIP-1α, MIP-1β) using a ProcartaPlex Mouse Basic kit (eBioscience). All secreted proteins were analyzed using the Bio-Plex 200 System/Bio-Plex Manager software, Version 6.0 (Bio-Rad, Mississauga, ON, Canada) according to the manufacturer’s instructions.

### 2.7. Cellular Protein Analysis

Protein was isolated from L6 myocytes (*n* = 7–8/treatment) treated with MCM for 24 h followed by stimulation with insulin (100 nM for 20 min) and used to assess the phosphorylation status of Akt signaling proteins IRS-1 (S636/S639), PTEN (S380), Akt (S473), GSK-3ab (S21/S9), and mTOR (S2448) using a Bio-Plex Pro Cell Signaling Akt Panel kit (Bio-Rad), with analysis using the Bio-Plex 200 System/Bio-Plex Manager software, Version 6.0 (Bio-Rad), according to the manufacturer’s instructions.

### 2.8. Glucose Uptake Assay

After 24 h treatment with MCM, L6 myocyte 2-deoxyglucose (2-DG) uptake was measured using a colorimetric Glucose Uptake Assay Kit (Abcam, Eugene, OR, USA), following the manufacturer’s instructions (*n* = 7–8/treatment). In brief, L6 myocytes were serum-reduced for 12 h followed by incubation with 100 mL of Krebs-Ringer-Phosphate-Hepes (KRPH) buffer containing 2% BSA for 40 min in a humidified incubator at 37 °C with 5% CO_2_. Cells were then stimulated with 1 µM insulin for 20 min, followed by the addition of 1 μM glucose analog, 2-DG, for 20 min. Through a series of reactions, 2-DG was converted to NADPH, leading to the production of an oxidized product that was detected by optical density (OD) at 412 nm using a spectrophotometer.

### 2.9. Statistical Analysis

All statistical analyses were performed using the Statistical Analysis System, University Edition (SAS Institute Inc., Cary, NC, USA), with *p* ≤ 0.05 considered statistically significant. Normal distribution was confirmed by the Shapiro–Wilk test. Data were analyzed by one-way ANOVA for the main effect of FA and followed, if justified, by a least-squared means post hoc test. Replicate experiments were averaged and expressed as means ± SEM.

## 3. Results

### 3.1. RAW 264.7 Fatty Acid Composition

As expected, AA and DHA and, correspondingly, *n*-6 and *n*-3 PUFAs were enriched in their respective treated macrophages compared to PA-treated macrophages (*p* < 0.0001 and *p* < 0.0001, respectively; Appendix A). Total PUFAs were increased in macrophages treated with DHA compared to PA- and AA-treated groups (*p* < 0.0001 and *p* = 0.0011, respectively) and in AA- compared to PA-treated groups (*p* < 0.0001; Appendix A). Interestingly, macrophages treated with AA and DHA had higher proportions of PA than the PA-treated macrophages (*p* = 0.0193 and *p* = 0.0012, respectively); however, the proportion of palmitoleic acid (18:1), the downstream metabolite of PA, was increased in PA-treated macrophages compared to AA- and DHA-treated macrophages (*p* < 0.0001 and *p* < 0.0001, respectively; Appendix A). Total SFA was increased in AA compared to PA (*p* = 0.0328), but there were no differences between the other FA treatments (*p* > 0.05; Appendix A).

### 3.2. mRNA Expression in RAW 264.7 Macrophages Treated with FA Followed by Stimulation with LPS

mRNA expression of M1 (*Tnfα, iNos, Cd11c*) and M2 (*Il-10, Tgfβ, Arg-1, Cd206*) polarization markers was measured in RAW 264.7 macrophages treated with FA for 24 h followed by stimulation with LPS for 12 h. Compared to PA and DHA, AA reduced mRNA expression of *Tnfα* (*p* = 0.005 and *p* = 0.0071, respectively; Figure 1A). AA and DHA decreased mRNA expression of *iNos* compared to PA (*p* < 0.0001 and *p* < 0.0001, respectively) but did not differ from each other (*p* > 0.05; Figure 1A). Further, *Cd11c* mRNA expression was higher in DHA-treated macrophages compared to PA and AA (*p* < 0.0001 and *p* = 0.0001, respectively), with no differences between AA and PA (*p* > 0.05; Figure 1A). mRNA expression of *Il-10* was increased by AA and DHA compared to PA (*p* < 0.0001 and *p* = 0.0004, respectively) and more potently by AA compared to DHA (*p* = 0.0204; Figure 1B). *Tgfβ* mRNA expression was increased by AA and DHA compared to PA (*p* = 0.0002 and *p* < 0.0001, respectively) and more potently by DHA compared to AA (*p* < 0.0001; Figure 1B). There was no difference in mRNA expression of *Arg-1* between FA treatments (*p* > 0.05); however, AA and DHA reduced expression of *Cd206* compared to PA (*p* < 0.0001 and *p* < 0.0001, respectively) but did not differ from each other (*p* > 0.05; Figure 1B).

### 3.3. Secreted Protein in RAW 264.7 Macrophages Treated with FA Followed by Stimulation with LPS

Inflammatory (TNFα, IL-6, MCP-1) and anti-inflammatory (IL-10) cytokines were measured in the supernatant of RAW 264.7 macrophages following 24 h treatment with FA and stimulation with LPS for 12 h. AA and DHA similarly reduced secreted TNFα (−12.8%, *p* = 0.0001 and −8.5%, *p* = 0.0086, respectively; Figure 2A), IL-6 (−74.3%, *p* < 0.0001 and −81.9%, *p* < 0.0001; Figure 2A) and MCP-1 (−85.6%, *p* < 0.0001 and −80.4%, *p* < 0.0001; Figure 2B) compared to PA; however, secreted levels of IL-10 did not differ between FA treatments (*p* > 0.05; Figure 2A).

### 3.4. NF-κB p65 and STAT3 Activation in RAW 264.7 Macrophages Treated with FA Followed by Stimulation with LPS

To determine if changes in inflammatory mediator production corresponded to changes in the activation of the NF-κB and STAT3 transcription factor complexes, phosphorylated NF-κB p65 and phosphorylated STAT3 were measured in the RAW 264.7 macrophages pre-treated with FA for 24 h followed by stimulation with LPS for 12 h. There were no differences in the ratios of phosphorylated NF-κB p65 (*p* > 0.05; Figure 3A) or STAT3 (*p* > 0.05; Figure 3B) between FA treatments.

### 3.5. mRNA Expression in L6 Myocytes Treated with MCM Derived from FA- and LPS-Stimulated RAW 264.7 Macrophages

mRNA expression of inflammatory (*Tnfα, Il-6, Il-1β, Mcp-1*) and anti-inflammatory (*Il-10*) cytokines was measured in L6 myocytes treated for 24 h with MCM derived from FA- and LPS-treated RAW 264.7 macrophages. There were no differences between MCM treatments in L6 myocyte mRNA expression of *Tnfα, Il-6, Il-1β,* or *Il-10* (*p* > 0.05; Figure 4A). However, AA- and DHA-derived MCM decreased mRNA expression of *Mcp-1* in L6 myocytes compared to PA (*p* < 0.0001 and *p* < 0.0001, respectively) but did not differ from each other (*p* > 0.05; Figure 4B).

### 3.6. Secreted Proteins from L6 Myocytes Treated for 24 h with MCM Derived from FA- and LPS-Treated RAW 264.7 Macrophages

Secreted inflammatory cytokines (TNFα, IL-6) and chemokines (MCP-1, MIP-1α, MIP-1β) were measured in L6 myocytes treated for 24 h with MCM derived from FA- and LPS-treated RAW 264.7 macrophages. Compared to PA, AA- and DHA-derived MCM similarly reduced L6 myocyte-secreted inflammatory cytokines TNFα (−99.4%, *p* < 0.0001 and −86.3%, *p* < 0.0001, respectively; Figure 5A) and IL-6 (−78.6%, *p* = 0.0066 and −80.0%, *p* = 0.0058, respectively; Figure 5A) as well as chemokines MCP-1 (−74.4%, *p* < 0.0001 and −74.7%, *p* < 0.0001, respectively) and MIP-1β (−99.6%, *p* < 0.0001 and −99.4%, *p* < 0.0001, respectively; Figure 5B). However, only DHA-derived MCM reduced the levels of the secreted chemokine, MIP-1α, compared to PA (−24.6%, *p* = 0.0004), with no difference compared to AA (*p* > 0.05; Figure 5B).

### 3.7. NF-κB p65 Activation in L6 Myocytes Treated with MCM Derived from FA- and LPS-Treated RAW 264.7 Macrophages

Phosphorylated NF-κB p65 was measured in the L6 myocytes treated for 24 h with MCM derived from FA- and LPS-treated RAW 264.7 macrophages. Compared to PA-derived MCM, AA and DHA tended to decrease phosphorylated NF-κB p65; however, the differences were not significant (*p* = 0.0761 and *p* = 0.082, respectively; Figure 6).

### 3.8. Phosphorylation of Akt Signaling Pathway Proteins in L6 Myocytes Treated with MCM Derived from RAW 264.7 Macrophages Stimulated with FA and LPS

Phosphorylation of positive (i.e., activation; Akt, GSK3ab, mTOR; Figure 7A) and negative (i.e., inactivation; IRS-1, PTEN; Figure 7B) regulators of the insulin signaling pathway were measured in L6 myocytes treated for 24 h with MCM derived from FA- and LPS- treated RAW 264.7 macrophages. AA increased phosphorylation of Akt (S473) compared to PA and DHA (46.6%, *p* < 0.0058 and 43.6%, *p* = 0.0082, respectively), which did not differ from each other (*p* > 0.05; Figure 7A). There were no differences between MCM treatment groups in other positive or negative regulators (*p* > 0.05; Figure 7A,B).

### 3.9. Insulin-Stimulated Glucose Uptake in L6 Myocytes Treated with MCM from RAW 264.7 Macrophages Stimulated with FA

Glucose uptake was measured in L6 myocytes treated for 24 h with MCM derived from FA- and LPS-treated RAW 264.7 macrophages. DHA-derived MCM increased myocyte glucose uptake by 16.5% compared to PA (*p* = 0.0417) but not compared to AA (*p* > 0.05), and AA did not differ from PA (*p* > 0.05; Figure 8).

## 4. Discussion

Obese skeletal muscle is characterized by the presence of M1 macrophages that contribute to the development of skeletal muscle inflammation and subsequent myocyte metabolic dysfunction [31]. Using an in vitro indirect co-culture model of myocytes cultured in conditioned media from macrophages stimulated with a physiological dose of LPS to mimic the obese skeletal muscle inflammatory microenvironment, this was the first study to investigate macrophage–myocyte interactions as a target for intervention with PUFA to mitigate skeletal muscle inflammation and ensuing insulin resistance. We provide evidence that the long-chain PUFA, AA, and DHA promote characteristics of macrophage polarization to the anti-inflammatory M2 phenotype and subsequently improve markers of myocyte insulin sensitivity.

In the current study, macrophages pre-treated with the long-chain *n*-3 PUFA, DHA, followed by stimulation with LPS had decreased M1 subset markers (*iNos* mRNA expression and TNFα, IL-6, and MCP-1 secreted protein) and increased M2 subset markers (*Il-10*, *Tgfβ,* and *Cd206* mRNA expression) compared to macrophages pre-treated with PA, an SFA. Our results are consistent with previous in vitro studies demonstrating that similar or even lower concentrations (50 μM) of DHA pre-treatment reduced M1 macrophage polarization markers stimulated by much higher doses of LPS (100 ng/mL) [19,20]. Specifically, DHA decreased macrophage expression [20] and secretion of TNFα and IL-6 [19] and increased secreted IL-10 [19] which, in contrast to the current study, was associated with reduced NF-κB activity [32]. Mechanistically, the anti-inflammatory effects of DHA have been shown through G coupled protein (GPR)120, wherein GPR120 ligand stimulation promotes the association of GPR120 with an adaptor protein, b-arrestin2 (barr2) [33]. The GPR120-barr2 complex is internalized and interferes with the TLR4/NF-κB signaling pathway [33]. Further, DHA has also been shown to disrupt the formation of lipid rafts necessary for TLR4 signaling, thereby further disrupting NF-κB signaling and activation [34]. Our study advances knowledge, since we used a physiological dose of LPS. Further study is required to understand the conditions under which NF-κB activation is altered, but it may be due to differences in the DHA and LPS doses and timing between studies.

The long-chain *n*-6 PUFA, AA, also decreased markers of M1 macrophage polarization (*Tnfα* and *iNos* mRNA expression and TNFα, IL-6, and MCP-1 secreted protein) and increased those of M2 polarization (*Il-10*, *Tgfβ,* and *Cd206* mRNA expression) compared to PA in the current study. Interestingly, the anti-inflammatory effects of AA in some measurements were more potent than that of DHA, demonstrated by a greater reduction in *Tnfα* and increase in *Il-10* mRNA expression compared to PA. These results were unexpected considering AA is metabolized by cyclooxygenases, lipoxygenases, and cytochrome P450 to generate metabolites that are generally considered inflammatory (e.g., eicosanoids) [35]. Indeed, our research group has shown that AA in the presence of LPS synergistically promotes inflammation in 3T3-L1 adipocytes by increasing *Mcp-1* and *Il-6* mRNA expression and secreted protein [25]. However, controversy exists, as AA-derived lipoxins are considered anti-inflammatory [36]. In macrophages, AA has been shown to support M2 polarization in a peroxisome proliferator-activated receptor (PPAR)-γ-dependent manner [22] and to reduce LPS-induced macrophage inflammation [35]. One suggested mechanism of the anti-inflammatory effects of AA independent of its downstream metabolites is the incorporation of AA within cellular membranes to improve membrane fluidity, which potentially modifies cellular signal transduction [37]. Another study also showed that low-dose AA (40 μM) directly inhibits PA-induced TLR4 signaling (and reduces TNFα and IL-6 production) by binding to its co-regulator, myeloid differentiation factor-2 (MD2) [35]. Further, treating macrophages with a higher dose of AA (250 μM) was shown to similarly counteract LPS-induced inflammatory cytokine production by disrupting the TLR4/NF-κB signaling cascade [24]. While previous work suggests TLR4/NF-κB signaling as potential mechanisms underlying the anti-inflammatory effects of AA in LPS-stimulated macrophages, more work is needed to confirm such mechanisms in our cell culture model and should further be verified in vivo. Collectively, our data support a role for both *n*-3 and *n*-6 PUFAs in modulating macrophage polarization towards the anti-inflammatory M2 phenotype.

While dietary *n*-3 and *n*-6 PUFAs have been shown to improve markers of skeletal muscle inflammation and insulin sensitivity [38,39], the current study is the first to examine the ability of *n*-3 and *n*-6 PUFAs to modulate macrophage-secreted cytokines and subsequently improve myocyte insulin sensitivity. In the current study, while both AA and DHA reduced macrophage-secreted inflammatory cytokines, only AA MCM increased myocyte phosphorylated Akt ^Ser473^, a marker of insulin sensitivity, and only DHA MCM increased myocyte glucose uptake compared to PA MCM. Obesity-associated inflammation is a suggested causal link to insulin resistance [1,2], as inflammatory cytokines have been shown to interfere with myocyte insulin signaling [40]. Similar experiments have demonstrated that purified inflammatory cytokines [8,9] or those derived from palmitate-treated macrophages [14] or inflammatory T helper 1 (Th1) polarized CD4^+^ T cells [5] reduce myocyte insulin sensitivity by reducing phosphorylation of Akt^Ser473^, leading to reduced glucose uptake. Mechanistically, TNFα and MCP-1 impair myocyte insulin signaling by triggering the IKK/NF-κB [41] and ERK1/2 [42] pathways, respectively, which in turn reduce Akt^Ser473^ phosphorylation and inhibit downstream insulin signaling and reduce expression of GLUT4. On the other hand, IL-6 is elevated in obesity and with exercise, and thus, its effects on insulin signaling are controversial [43]. Myocytes treated with IL-6 demonstrate improved insulin signaling and glucose uptake [44,45], while both chronically elevated IL-6 and IL-6 infusion in mice have shown reduced insulin-stimulated glucose uptake [46]. In contrast to our findings, the *n*-3 PUFA had no effect on modulating secreted inflammatory cytokines in conditioned media derived from insulin resistant 3T3-L1 adipocytes or the ex vivo AT of high-fat diet-fed rats, which subsequently led to no changes in myocyte insulin sensitivity [47,48]. Therefore, our data suggest that *n*-3 and *n*-6 PUFA-induced improvement in myocyte insulin sensitivity may be driven, in part, by the reduction of inflammatory cytokines, confirming that macrophage–myocyte cross-talk is a promising target for intervention in this context. However, future work should confirm this as well as investigate other cells and tissues involved in myocyte cross-talk in vivo that contribute to obesity-induced skeletal muscle insulin resistance.

Myocytes respond to inflammatory stimuli by producing mediators that further perpetuate the inflammatory microenvironment in obese skeletal muscle [49]. Indeed, secreted mediators from palmitate-treated macrophages have been shown to increase myocyte mRNA expression of inflammatory cytokines and chemokines, including *Tnfα* and *Mcp-1* [14,50]. In the current study, there was a substantial increase in secreted TNFα, IL-6, and MCP-1 in MCM after being added to L6 myocytes for 24 h. While dietary *n*-3 PUFAs have been shown to reduce obese skeletal muscle inflammation [16], this is the first study to show that *n*-3 and *n*-6 PUFA-induced changes in macrophage-derived factors reduced myocyte inflammation. Specifically, AA and DHA decreased myocyte mRNA expression of MCP-1 and myocyte-secreted inflammatory cytokines and chemokines (TNFα, IL-6, MCP-1, and MIP-1β), which corresponded with a trend towards reduced NF-κB phosphorylation in L6 myocytes. Like macrophages, myocytes also express TLR4, and therefore, respond to SFA and LPS by increasing the production of inflammatory cytokines and chemokines [50,51]. Increased secreted chemokines such as MCP-1 promote the infiltration of more macrophages, which contributes to a vicious cycle, perpetuating skeletal muscle inflammation in obesity [12]. Overall, our results suggest that AA- and DHA-induced reduction in macrophage-secreted cytokines subsequently supports attenuated inflammation in myocytes to mitigate inflammatory macrophage–myocyte cross-talk and ensuing myocyte IR.

A key strength of the current study was the utilization of physiological doses of FA and LPS to mimic obese in vivo conditions in our in vitro co-culture model. In humans, plasma PUFA levels vary, but 100 μM has been reported for DHA and even higher levels for AA; thus our dose is appropriate and achievable through diet [52,53]. Similarly, our 10 ng/mL dose of LPS corresponds to the 5 endotoxin units (EU)/mL reported in obese rodents [27,28] and humans [26]. However, our use of cell lines instead of primary cells reduces some of the physiological relevance. Further, using an indirect co-culture model where RAW 264.7 macrophages and L6 myocytes were cultured individually allowed us to determine the source of macrophage and myocyte mRNA expression and macrophage-secreted cytokines that would interact with myocytes in vitro. However, due to the presence of macrophage-derived cytokines in MCM, we were unable to specify the L6 myocyte-secreted cytokines. Nonetheless, there was an increase in secreted cytokines after treating L6 myocytes with MCM, and the differences between treatment groups were maintained. Further, since macrophages and myocytes are in direct contact in vivo, we may be missing some contact-dependent effects, as demonstrated previously [50]; hence future studies should explore the effects of *n*-6 and *n*-3 PUFAs in attenuating contact-dependent macrophage–myocyte crosstalk and the role of myocyte-secreted cytokines in macrophage polarization.

## 5. Conclusions

In conclusion, our results suggest that the long-chain PUFAs AA and DHA modulate macrophage polarization and subsequent cross-talk with myocytes, ultimately improving myocyte insulin signaling and glucose uptake. Collectively, our work supports dietary long-chain *n*-3 and *n*-6 PUFAs as a strategy to mitigate inflammatory and metabolic dysfunction in obesity.

## Figures and Tables

**Figure 1 nutrients-16-02086-f001:**
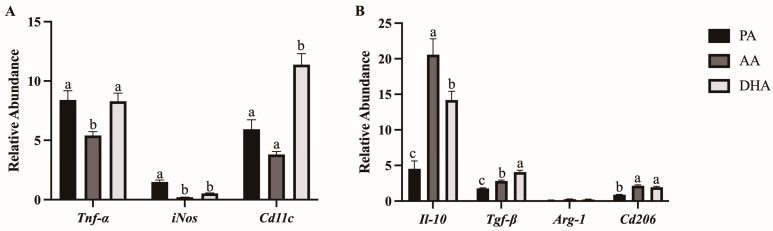
mRNA expression for markers of (**A**) M1 and (**B**) M2 macrophage polarization in RAW264.7 macrophages treated with 100 μM total FA for 24 h followed by 10 ng/mL LPS for 12 h. Values are means ± SEM; *n* = 7/treatment. Data were normalized to 18S mRNA expression and presented as relative abundance and analyzed by one-way ANOVA for the effect of FA treatment. Means without a common letter are significantly different, *p* < 0.05.

**Figure 2 nutrients-16-02086-f002:**
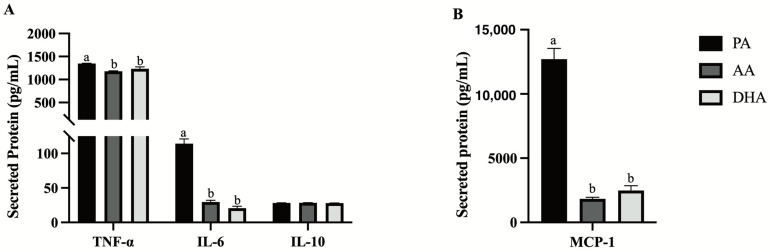
Secreted (**A**) cytokines and (**B**) MCP-1 (chemokine) from RAW264.7 macrophages treated with 100 μM total FA for 24 h followed by 10 ng/mL LPS for 12 h. Values are means ± SEM; *n* = 7–9/treatment. Data were analyzed by one-way ANOVA for the effect of FA treatment. Means without a common letter are significantly different, *p* < 0.05.

**Figure 3 nutrients-16-02086-f003:**
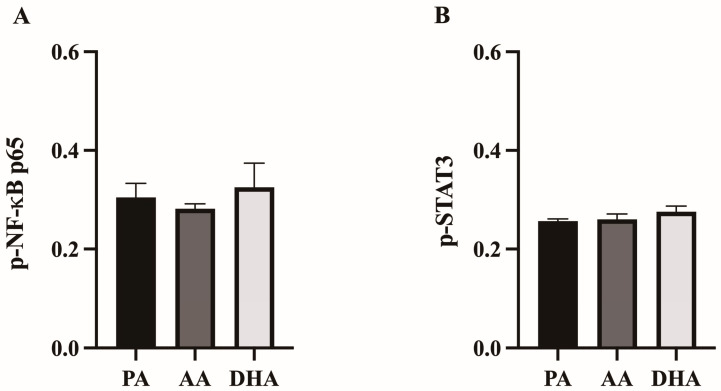
Ratio of phosphorylated (**A**) NF-κB and (**B**) STAT3 in RAW264.7 macrophages treated with 100 μM total FA for 24 h followed by 10 ng/mL LPS for 12 h. Values are means ± SEM; *n* = 6–7/treatment. Data were analyzed by one-way ANOVA. There were no significant differences, *p* > 0.05.

**Figure 4 nutrients-16-02086-f004:**
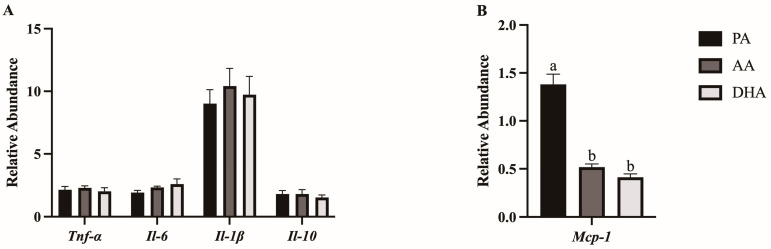
mRNA expression of (**A**) cytokines and (**B**) chemokines in L6 myotubes treated for 24 h with MCM derived from RAW264.7 macrophages treated with 100 μM total FA for 24 h followed by 10 ng/mL LPS for 12 h. Values are means ± SEM; *n* = 5–6/treatment. Data were normalized to 18S mRNA expression and are presented as relative abundance and analyzed by one-way ANOVA for the effect of FA treatment. Means without a common letter are significantly different, *p* < 0.05.

**Figure 5 nutrients-16-02086-f005:**
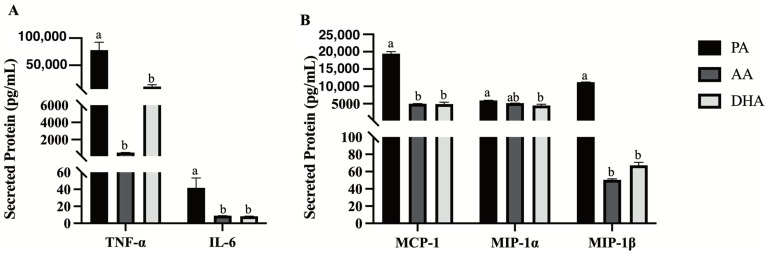
Secreted (**A**) cytokines and (**B**) chemokines of combined macrophage and L6 myotubes in supernatant from L6 myotubes treated for 24 h with MCM derived from RAW264.7 macrophages treated with 100 μM total FA for 24 h followed by 10 ng/mL LPS for 12 h. Values are means ± SEM; *n* = 7–8/treatment. Data were analyzed by one-way ANOVA for the effect of FA treatment. Means without a common letter are significantly different, *p* < 0.05.

**Figure 6 nutrients-16-02086-f006:**
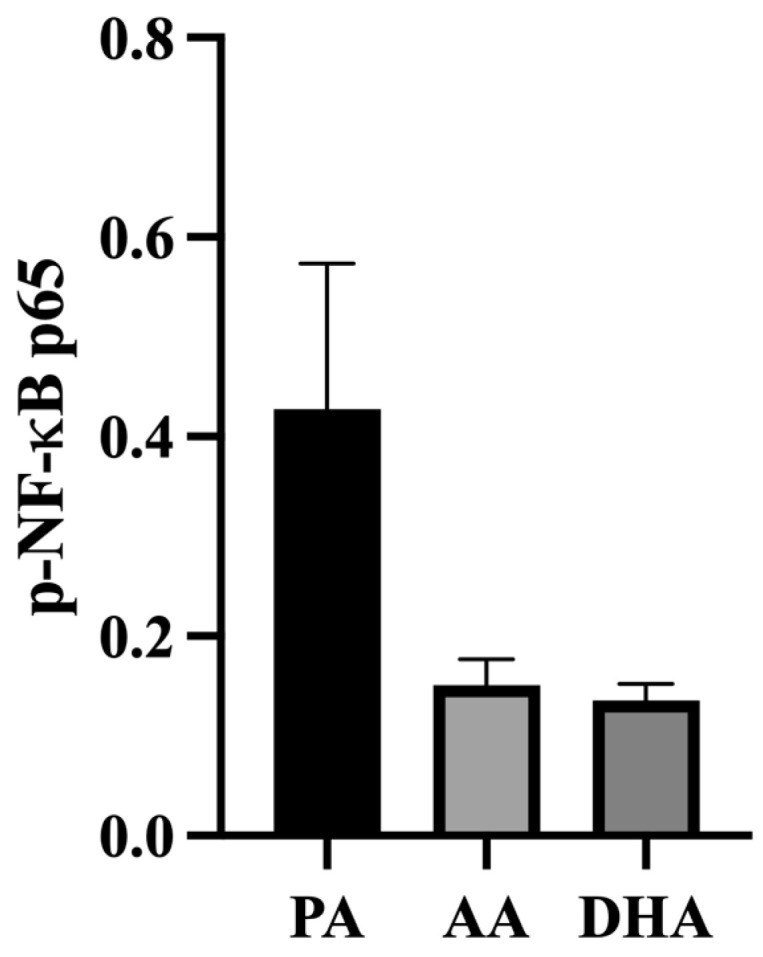
Phosphorylated NF-κB in L6 myotubes treated for 24 h with MCM derived from RAW264.7 macrophages treated with 100 μM total FA for 24 h followed by 10 ng/mL LPS for 12 h. Values are means ± SEM; *n* = 6–8/treatment. Data were analyzed by one-way ANOVA. There were no significant differences, *p* > 0.05.

**Figure 7 nutrients-16-02086-f007:**
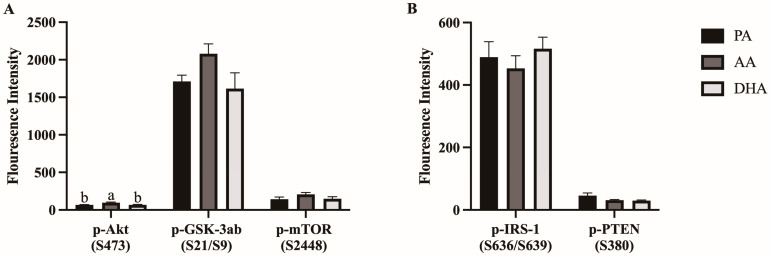
Phosphorylation of (**A**) positive and (**B**) negative regulators of the insulin signaling pathway measured in L6 myocytes stimulated with 100 nM insulin for 20 min after being treated for 24 h with MCM derived from RAW264.7 macrophages treated with 100 μM total FA for 24 h followed by 10 ng/mL LPS for 12 h. Values are means ± SEM; *n* = 7–8/treatment. Data were analyzed by one-way ANOVA for the effect of FA treatment. Means without a common letter are significantly different, *p < 0.05*.

**Figure 8 nutrients-16-02086-f008:**
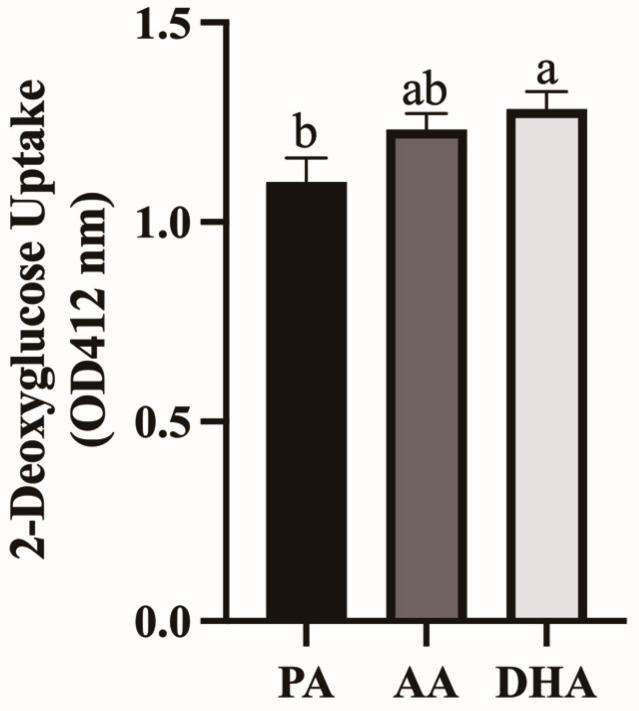
2-deoxyglucose uptake measured in L6 myocytes stimulated with 100 nM insulin for 20 min after being treated for 24 h with MCM derived from RAW264.7 macrophages treated with 100 μM total FA for 24 h followed by 10 ng/mL LPS for 12 h. Values are means ± SEM; *n* = 7–8/treatment. Data were analyzed by one-way ANOVA for the effect of FA treatment. Means without a common letter are significantly different, *p < 0.05*.

## Data Availability

Data are contained within the article. The data presented in this study are available upon reasonable request sent to the corresponding author.

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
