# Peer review of "n-3 and n-6 Polyunsaturated Fatty Acids Modulate Macrophage–Myocyte Inflammatory Crosstalk and Improve Myocyte Insulin Sensitivity"

_nutrients, 2024, doi:10.3390/nu16132086_

Round 1
Reviewer 1 Report
Comments and Suggestions for Authors
Manuscript is interesting and well written, addressing the cellular and molecular mechanisms governing the macrophage-myocyte interactions as a target for PUFAs to mitigate skeletal muscle inflammation and improving insulin resistance. For this purpose, an in vitro indirect co-culture model of myocytes cultured in conditioned media from macrophages stimulated with a physiological dose of LPS, was used to mimic the obese skeletal muscle inflammatory microenvironment, and in this setting the effects of PA, AA and DHA were evaluated. Interestingly, the effects induced by omega 3 and omega 6 acids are not opposing, and also, the anti-inflammatory effects of AA in some assays were more potent than those of DHA .
Author Response
Thank you for reviewing our work and for your considerate and positive comments. I didn't see any specific changes to be made, but please do let me know if I have missed something here and I will be very happy to address any specific comments. Thank you again, Lindsay
Reviewer 2 Report
Comments and Suggestions for Authors
This study determined the effects of the long-chain n-6 Polyunsaturated Fatty Acids (PUFA), arachidonic acid (AA), and n-3 PUFA, docosahexaenoic acid (DHA), on macrophage M1 polarization and subsequent effects on myocyte inflammation and metabolic function compared to the saturated fatty acids (SFA) and palmitic acid (PA) using an in vitro model that mimics the obese skeletal muscle inflammatory microenvironment.
The objective and conclusion in the Abstract need to be written clearer.
Figure 1, 2, 4, 5, 6, 8. Please clarify what the letter above the column bar (a, b, c, ab) mean in the Figure legend.
Line 367, Please replace word” In summary” with the word “ Conclusion”.
Author Response
Comment 1: The objective and conclusion in the Abstract need to be written clearer.
Response 1: Thank you for this comment. We are pleased to address this as it will improve the readability of our work. We have highlighted the changes in yellow in the revised manuscript Abstract.
Comment 2: Figure 1, 2, 4, 5, 6, 8. Please clarify what the letter above the column bar (a, b, c, ab) mean in the Figure legend.
Response 2: Thank you for this comment that will help to clarify our findings to the reader. In the figure legend under the relevant figures, we have changed the wording from "Means without a common letter differ, P<0.05." to "Means without a common letter are significantly different, P<0.05." This was done for Figures 1,2,4,5,7 and 8 (but not Figures 3 and 6 as there were no significant differences and thus no letters above the column bars).
Comment 3: Line 367, Please replace word” In summary” with the word “ Conclusion”.
Response 3: Thank you for this change. We have adjusted this in the revised manuscript on line 367 and highlighted in yellow.